# Purging due to self-fertilization does not prevent accumulation of expansion load

**Leo Zeitler**[ID]*, **Christian Parisod**[ID], **Kimberly J. Gilbert**[ID]*

Department of Biology, University of Fribourg, Fribourg, Switzerland

* leo.zeitler@unifr.ch (LZ); kimberly.gilbert@unifr.ch (KJG)

## Abstract

As species expand their geographic ranges, colonizing populations face novel ecological conditions, such as new environments and limited mates, and suffer from evolutionary consequences of demographic change through bottlenecks and mutation load accumulation. Self-fertilization is often observed at species range edges and, in addition to countering the lack of mates, is hypothesized as an evolutionary advantage against load accumulation through increased homozygosity and purging. We study how selfing impacts the accumulation of genetic load during range expansion via purging and/or speed of colonization. Using simulations, we disentangle inbreeding effects due to demography versus due to selfing and find that selfers expand faster, but still accumulate load, regardless of mating system. The severity of variants contributing to this load, however, differs across mating system: higher selfing rates purge large-effect recessive variants leaving a burden of smaller-effect alleles. We compare these predictions to the mixed-mating plant *Arabis alpina*, using whole-genome sequences from refugial outcrossing populations versus expanded selfing populations. Empirical results indicate accumulation of expansion load along with evidence of purging in selfing populations, concordant with our simulations, suggesting that while purging is a benefit of selfing evolving during range expansions, it is not sufficient to prevent load accumulation due to range expansion.

**Data Availability Statement:** Genetic data is archived at NCBI SRA (accession PRJNA773763). Code and simulation output is available on GitHub at https://github.com/LZeitler/selfing_expansion.

## Author summary

The geographic space that species occupy, i.e., the species range, is known to fluctuate over time due to changing environmental conditions. Since the most recent glaciation, many species have recolonized available habitat as the ice sheets melted, expanding their range. When populations at species range margins expand into newly available space, they suffer from an accumulation of deleterious alleles due to repeated founder effects. We study whether self-fertilization, which is considered an evolutionary dead-end, can be favored under these expanding edge conditions. Selfing has two important effects: allowing for faster expansion due to reproductive assurance and purging recessive deleterious alleles by exposing them to selection as homozygotes. We use simulations to identify the impact of selfing on expanded populations and then compare these results to an empirical dataset to assess whether our predictions are met. We use the mixed-mating plant alpine

**Funding:** This research was funded by Swiss National Science Foundation Ambizione grant #PZ00P3_185952 to K.J.G. L.Z. and K.J.G. both received salary from this funding source. The funders had no role in study design, data collection and analysis, decision to publish, or preparation of the manuscript.

**Competing interests:** The authors have declared that no competing interests exist.

rock-cress (*Arabis alpina*) since it has both expanded since the last glaciation and undergone a mating shift to selfing. We find that selfing does not prevent the accumulation of deleterious load, however purging does still act to remove the most severe variants, indicating that selfing provides this benefit during range expansions.

## Introduction

Species across the globe have expanded or shifted their species ranges in response to changing climates [1, 2]. Among such range expansions are multitudes of plant species which underwent expansions when recolonizing after the last glacial maximum. An interesting aspect of plant range expansions is the widely observed feature that many plant species exhibit a transition to self-fertilization ('selfing') at range edges [3, 4]. Selfing is often considered to be an evolutionary dead end, as it can lead to mutational meltdown, transitions back to outcrossing are not observed, and extinction in selfing clades is greater than diversification [5–9]. The observation of enrichment for selfing at species range edges is thus surprising, leading to the hypothesis that the demographic and evolutionary conditions that range expansions impose may convey advantages to this mating system.

Range expansions create unique evolutionary and demographic conditions through repeated founder events and population bottlenecks as individuals colonize new territory. Small colonizing populations are subject to reduced efficiency of selection and increased strength of genetic drift [10, 11] which results in the process of gene surfing, whereby variants increase in frequency at expanding edges due to serial founder events [12–15]. Accumulation of deleterious variants in expanded edge populations due to gene surfing has been evidenced as creating what is termed expansion load [16–18]. This expansion load has the potential to temporarily halt population growth or cause local extinction at the boundaries of the species range [16, 17, 19, 20], creating a significant evolutionary challenge for adaptation and survival during range expansion. On top of these evolutionary challenges, individuals colonizing previously unoccupied environments also face ecological challenges such as Allee effects due to reduced population sizes, often manifesting as limited mate (or pollinator) availability and therefore further slowing colonization and expansion [21–25].

In terms of the evolutionary challenge presented during range expansions, previous theoretical work has well established expectations for the accumulation of expansion load [12–20]. Empirical evidence of expansion load has also been documented in systems such as humans [26–28], plants [29, 30], and experimental bacterial populations [15, 31], with new studies continuing to emerge. Yet, how the evolution of self-fertilization at range edges may impact the dynamics of expansion load accumulation is not well understood. Empirical evidence in *Mercurialis annua* [32] and theoretical results from simulations [33] suggest that range expansions may facilitate a transition to selfing by depleting the genetic load at the edge and reducing inbreeding depression. Other empirical studies have identified load in expanded populations [30] and determined that variation in selfing rate across a species range does not correspond to pollinator availability [34]. What remains to be understood, and what we investigate in this study, is exactly how much of an evolutionary advantage purging by selfing may provide during range expansions: what are the dynamics and characteristics of load accumulation and purging under different selfing rates, and can purging fully prevent expansion load?

Selfing increases homozygosity which can express inbreeding depression and reduce fitness [35–37], but selfing can alternatively also purge recessive deleterious variants and increase fitness [33, 38–44]. We can thus hypothesize that if purging is a significant factor, it may serve as

an evolutionary advantage during range expansion and contribute to the observation of increased selfing at species range edges. Glémin [43] concluded that purging might only be a short term effect of selfing, and over long evolutionary time scales fixation would be the major outcome in selfers. Furthermore, evidence suggests that purging by selfing in small populations is less feasible, and small-effect deleterious variants can still contribute to an increase in genetic load [45]. Because selfing has several related effects beyond purging, it is necessary to disentangle these to try to understand what evolutionary benefits selfing may convey. Two evolutionary impacts of selfing are a reduction in effective population size and a reduction in the effective recombination rate [46–48], and a third, albeit ecological, impact of selfing is to remove or reduce Allee effects. All of these effects are likely to play a role during species range expansions. Selfing removes Allee effects by assuring reproductive success in low density populations [49, 50], therefore leading to faster expansion speeds, as already evidenced by some studies of organisms with uniparental reproduction [51, 52]. The reduction in $N_e$ along with reduced effective recombination rates should both be disadvantageous and exacerbated when compounded with the already reduced genetic diversity due to founder effects during range expansions.

Our current study aims to understand if and how purging through selfing serves as an evolutionary advantage during range expansion by removing accumulated load. Combining theoretical approaches with an empirical investigation, we provide insight into the underlying dynamics of load accumulation and evidence of observable signatures in natural populations. Using simulations, we investigate these dynamics through time as well as within different classes of deleterious variants, so that we can compare across a range of selfing rates and other mutational parameters. Forward-time, individual-based simulations of a range expansion with or without a mating system shift provide a full understanding of the dynamics of load accumulation in the presence or absence of selfing. We directly assess outcomes across differing values of dominance coefficients and shapes of the distribution of mutational effects, including lethal, mildly deleterious, and beneficial variants, for a complete understanding of the signature that purging leaves in the genome. We also compare how these load accumulation and purging dynamics differ over different rates of evolved selfing. Since it is known from previous theoretical work that the speed of range expansion plays a role in the severity of load accumulation [20], we additionally report on how selfing impacts the speed of range expansion in our simulations through reproductive assurance.

We qualitatively test the predictions from our simulations in a relevant empirical system: the perennial arctic-alpine plant *Arabis alpina* L. This species is known to have been subject to range expansions and contractions in response to the repeated quaternary climate oscillations [53, 54], providing a useful system in which to test our simulation predictions. The Italian peninsula is a known refugium for outcrossing populations during the last glacial maximum [54]. Post-glaciation, the species then recolonized alpine habitats across Europe and concurrently with this expansion evolved a mating system of predominant self-fertilization [54–56]. Italian populations of *A. alpina* are predominantly outcrossing with high genetic diversity [54, 56–58], while in the French and Swiss Alps populations are mainly selfing, with higher homozygosity and lower nucleotide diversity [54, 56, 57, 59]. We investigate signatures of load accumulation and purging across these populations of *A. alpina* and discuss how this relates to our simulation results.

Combining theoretical and empirical investigations in evolutionary biology provides useful, albeit still limited, assessments of why and how nature matches expectations from theory. Our study thus also highlights where the most valuable advances in empirical biology may be useful. Estimating fitness in natural populations remains a difficult and time-consuming task, but proxies and inference methods for mutation load have become increasingly used in the

evolutionary biology literature. In this study, our analyses across both simulated and empirical data of load accumulation, distribution of fitness effects, and changes in genetic diversity give insight into the evolution of selfing and why range expansions may favor such a transition. We find that selfing does not prevent the accumulation of deleterious load, however purging does still act to remove the most severe variants, indicating that selfing provides a benefit through purging during range expansions and furthermore changes the distribution of how load is realized across the genome.

## Results

### Selfing leads to faster expansion

Using individual-based, forward-time simulations in SLiM v3.7.1 [60], we modelled a range expansion across a one-dimensional linear landscape. We simulated obligate outcrossing from the first deme ('core') followed by a shift to self-fertilization in the 25th deme of the landscape (out of 50 total demes), with selfing rates $\sigma$ of 0.5, 0.95, or 1, or for a null comparison, simulations with continued obligate outcrossing across the entire landscape during expansion and colonization (Fig 1A).

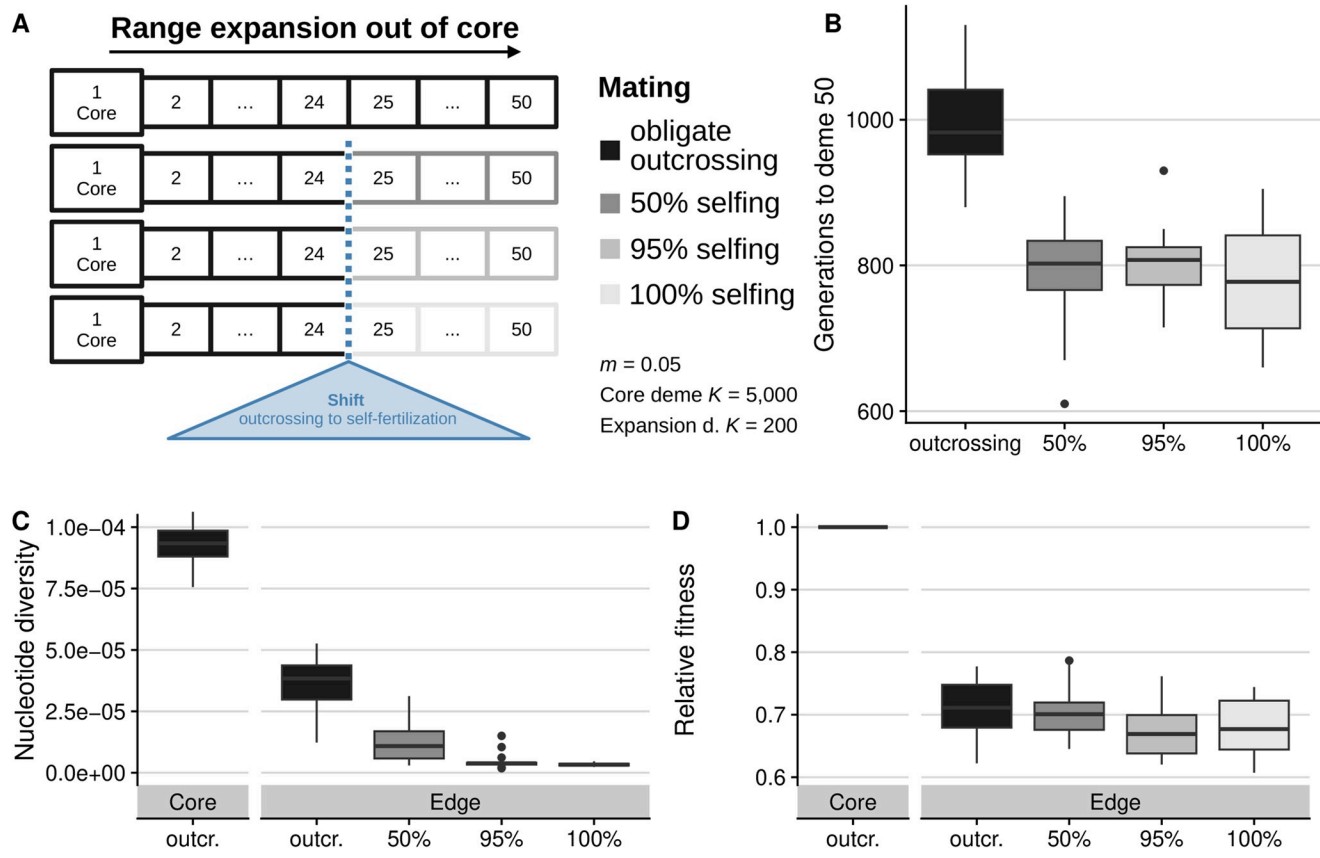

**Fig 1.** Simulation schematic for 1-D landscapes with stepping-stone migration (A). A shift in the rate of self-fertilization occurs in the center of the landscape (blue triangle). The number of generations needed to cross the landscape from the core to deme 50 (B). Expansion time was lower for all selfing rates but higher for obligate outcrossing. Mean nucleotide diversity (C) was reduced outside of the core, with a greater reduction for higher selfing: $\overline{\pi}_{core} = 9.235 \times 10^{-5}, \overline{\pi}_{\sigma=0} = 3.646 \times 10^{-5}, \overline{\pi}_{\sigma=0.5} = 1.274 \times 10^{-5}, \overline{\pi}_{\sigma=0.95} = 4.608 \times 10^{-6}, \overline{\pi}_{\sigma=1} = 3.410 \times 10^{-6}$. Relative fitness (D) decreased from core to edge with similar values across outcrossers and selfers: $\overline{\omega}_{\sigma=0} = 0.709, \overline{\omega}_{\sigma=0.5} = 0.702, \overline{\omega}_{\sigma=0.95} = 0.676, \overline{\omega}_{\sigma=1} = 0.683$, reflecting a relative loss of fitness as compared to the core of 29.1%, 29.8%, 32.4%, and 31.7% respectively for $\sigma = 0, 0.5, 0.95$, and 1.

One expected benefit of selfing during range expansion is increased expansion speed, which we observed in our simulation results. We compared the number of generations required to cross the landscape among selfing rates and found that mixed mating and obligate selfing populations had a faster expansion speed compared to obligate outcrossing populations (see Fig 1B). The differences in expansion time among different selfing rates was minor (mean expansion times in generations for $\sigma$ = 0.5, 0.95, and 1, respectively: 792 ($SD$ = 78.5), 800 ($SD$ = 51.4), 777 ($SD$ = 70.7)) compared to the notable difference in expansion time of obligate outcrossers (990 generations ($SD$ = 69.8)).

## Range expansion increases genetic load and decreases diversity in simulations

To test if and how selfing modifies the outcomes of a range expansion in our simulations, we examined genetic diversity in expanded populations and across selfing rates. Both outcrossing and selfing edge populations showed large reductions in diversity due to the expansion. Outcrossers retained the highest nucleotide diversity for neutral sites ($\pi_{edge,\sigma=0}$ = 3.646 × 10$^{-5}$), while with increasing self-fertilization rates, populations showed further reductions in nucleotide diversity ($\pi_{edge,\sigma=1}$ = 3.410 × 10$^{-6}$, Fig 1C). Core populations which never experienced expansion and always outcrossed had the highest nucleotide diversity ($\pi_{core}$ = 9.235 × 10$^{-5}$).

Genetic load is predicted to be higher in expanded populations, so we next examined how selfing modulates this outcome of a range expansion. With simulations we could accurately distinguish inbreeding effects due to mating of related individuals in small populations at the range edge versus inbreeding effects resulting from uni-parental inheritance, i.e., selfing, by contrasting obligate outcrossing scenarios to those with various rates of selfing. Fitness of every individual is also known, as this is defined in SLiM as the target number of offspring to be generated by an individual and is calculated multiplicatively across the effects of all derived mutations (see Methods for a full description). We calculated mean fitness of all individuals within a deme per replicate ($\overline{\omega}$) and compared these values between core and edge populations after the simulated expansion was complete. In all cases, the range expansion reduced fitness at the edge due to expansion load (Fig 1D). In the obligate outcrossing case ($\sigma$ = 0), we observed a reduction of fitness from core to edge of 29.1%. Interestingly, selfers showed negligible differences in load accumulation relative to outcrossers, with at most a mean reduction in fitness of 31.7% for obligate selfers. The same qualitative results were observed for additional simulation parameter sets that tested the sensitivity of our results to mutational parameters (see S1 Fig). We observed an increase in the proportion of loci fixed for deleterious alleles in all expanded populations, with these proportions increasing for higher selfing rates (S2(A) Fig). Similarly, we found that mean counts of deleterious loci (recessive model) increased from core to edge as well as from lower to higher rates of selfing (S2(B) Fig), whereas mean counts of deleterious alleles (additive model) showed less to no clear pattern from core to edge and among selfing rates (S2(C) Fig).

To understand why and how self-fertilization seemingly had no impact on removing genetic load during a range expansion, we examined demes over time and space to disentangle the effects of inbreeding due to demography versus inbreeding due to increased self-fertilization (Fig 2). Outcrossing deme 24 exhibited a mean observed heterozygosity level of $\overline{H}_{24}$ = 1.47 × 10$^{-4}$ when it was first colonized during the expansion, i.e., when it was the edge of the species range. Beyond deme 24 the mating system shifts to selfing and we observed a continual loss of heterozygosity at the expanding front. When the expansion front reached deme 35, $\overline{H}_{35}$ for 95% selfers rapidly decreased to 7.05 × 10$^{-7}$. For outcrossers, however, heterozygosity exhibited a more gradual rate of reduction across the course of expansion, reaching $\overline{H}_{35}$ = 1.19 × 10$^{-4}$ by the time deme 35 was colonized (Fig 2A and S3

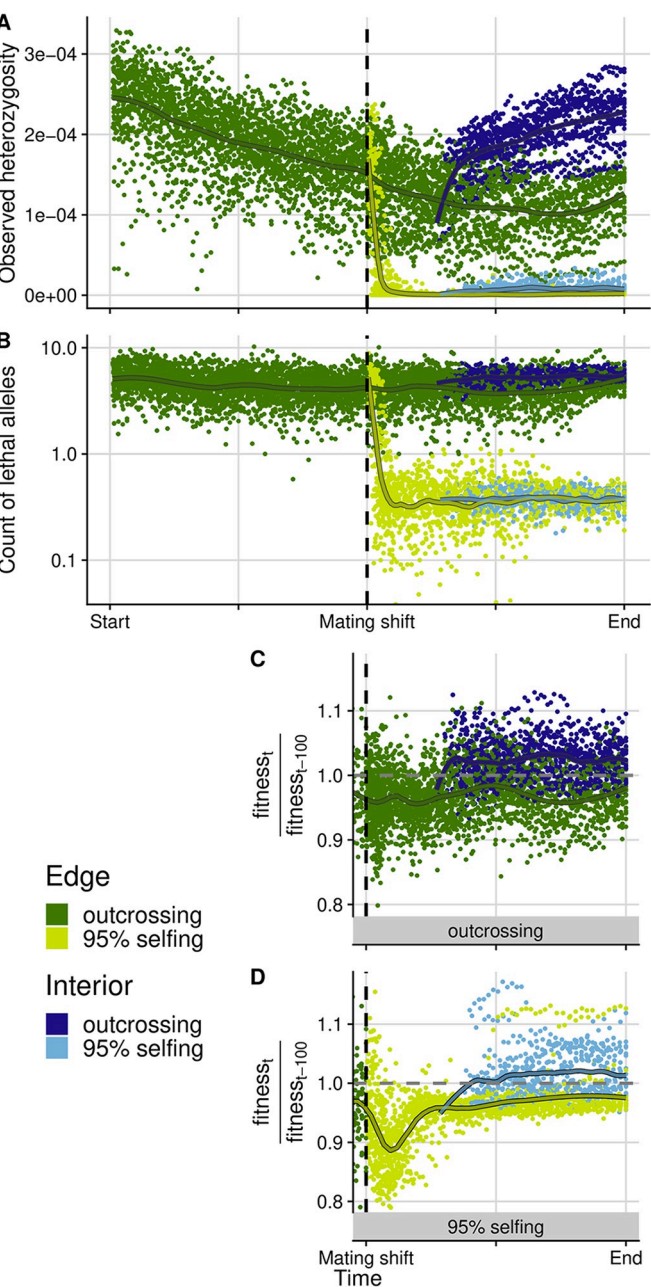

**Fig 2.** Observed heterozygosity (A), count of lethal alleles (B), and the rate of fitness change (C-D) are shown through time for the expanding range edge (green) as compared to change over time within one interior deme, stationary on the landscape (blue, deme 35). For selfing rates 50% and 100%, see S3 and S4 Figs. Panels (C) and (D) show the rate of fitness change as measured over 100-generation intervals, separately for outcrossers and selfers. A value of 1 indicates no change in fitness over 100 generations while values above 1 indicate increasing fitness and values below 1 indicate fitness loss. The vertical dashed line indicates the point in time where the mating system shifts to selfing. This shift occurs at deme 25 on the landscape, and since there is variation across simulation replicates in the generation time taken to expand to deme 25, we plotted all values relative to this time point for each replicate shown ($n$ = 20 replicates per selfing rate scenario). Each point is the value from a single simulation replicate and lines are loess (span = 0.2) fitted curves across all replicates.

Fig). We examined how diversity recovered in deme 35 over time since its colonization until the end of the simulation and found that outcrossers recovered to higher levels ($\overline{H}_{35}$ at the end of the simulation $2.27 \times 10^{-4}$) than selfers (for 95% selfing $\overline{H}_{35} = 7.72 \times 10^{-6}$).

**Fitness loss despite genetic purging.** We counted the number of lethal alleles per individual and observed a reduction in the count of lethals that corresponds with the reduction in heterozygosity (Fig 2A and 2B). Lethal alleles were only reduced when the shift to selfing occurred, and we observed the same pattern at every simulated selfing rate. Obligate outcrossers did not exhibit a reduction in lethal alleles, showing no evidence of purging. The largest reduction in lethal alleles occurred for the shift to the highest selfing rate ($\sigma = 1$) with a 91.69% drop in lethal alleles, while our lowest simulated selfing rate ($\sigma = 0.5$) still showed a strong effect of purging lethal alleles with an 83.74% reduction (S4 Fig).

The rate of change of fitness in a given deme, measured at a focal generation ($t$) compared to 100 generations prior ($t - 100$), showed a consistent loss of fitness over time due to range expansion as well as some fitness recovery in populations behind the expanding front (Fig 2C and 2D). Edge demes which recently underwent the shift to selfing exhibited a drastic reduction in fitness relative to equivalent outcrossers. This high rate of fitness loss exhibited by selfers is temporary and only lasts for between 45–235 generations, after which the rate of fitness loss recovers to the same rate as that observed in expanding obligate outcrossers: still below one on average and accumulating expansion load.

We also investigated the impact the range expansion and mating shift had on the realized distribution of selection coefficients. Overall, we found the greatest proportion of deleterious mutations in the weak to intermediate bin of selection coefficients ($-0.0001 \leq s < -0.001$), with just below 60% of all sites falling into this class (Fig 3). The next most deleterious bin ($-0.001 \leq s < -0.01$) contained about 30% of sites, while about 10% of sites are in the weakest selection coefficient bin. Lethal alleles made up a small proportion of segregating sites, as expected given the small proportion defined in the simulation parameters. Within these small numbers of severely deleterious variants, there was a consistent trend for a reduction of lethals from core to edge of nearly 50% for outcrossers and significantly further reduction for all rates of selfing, increasing from a nearly 75% reduction from core to edge for 50% selfers to more than 75% for 100% selfers (Fig 3 inset). In our additional simulated parameter sets, this qualitative pattern of reduction in lethals also holds in all cases, with some quantitative differences across the different DFE shapes and across the different dominance parameters (S5 Fig). The pattern of reduced proportions of deleterious sites as selfing rate increases holds in both the lethal category as well as the second-most deleterious allele class. We consistently observed the reverse pattern in the remaining weaker effect bins, with proportions of weakly deleterious sites slightly increasing at range edges and more so with higher selfing rates. This observation is consistent with more efficient removal of highly deleterious alleles and mutation accumulation at sites with smaller absolute selection coefficients.

## Reduced genetic diversity and elevated load in expanded selfing *A. alpina* populations

To test if our observations from simulations for genetic diversity, load accumulation, and purging are realized in natural populations, we used the mixed mating plant *A. alpina*, which underwent a range expansion concurrently with a shift to higher self-fertilization rates from Italy (outcrossing) into the Alps (selfing, [54, 56]; see Fig 4A). Using 191 newly sampled and sequenced short-read genomes from Italy and France combined with publicly available data from Switzerland and across Europe [58, 61], we examined differences across the species range

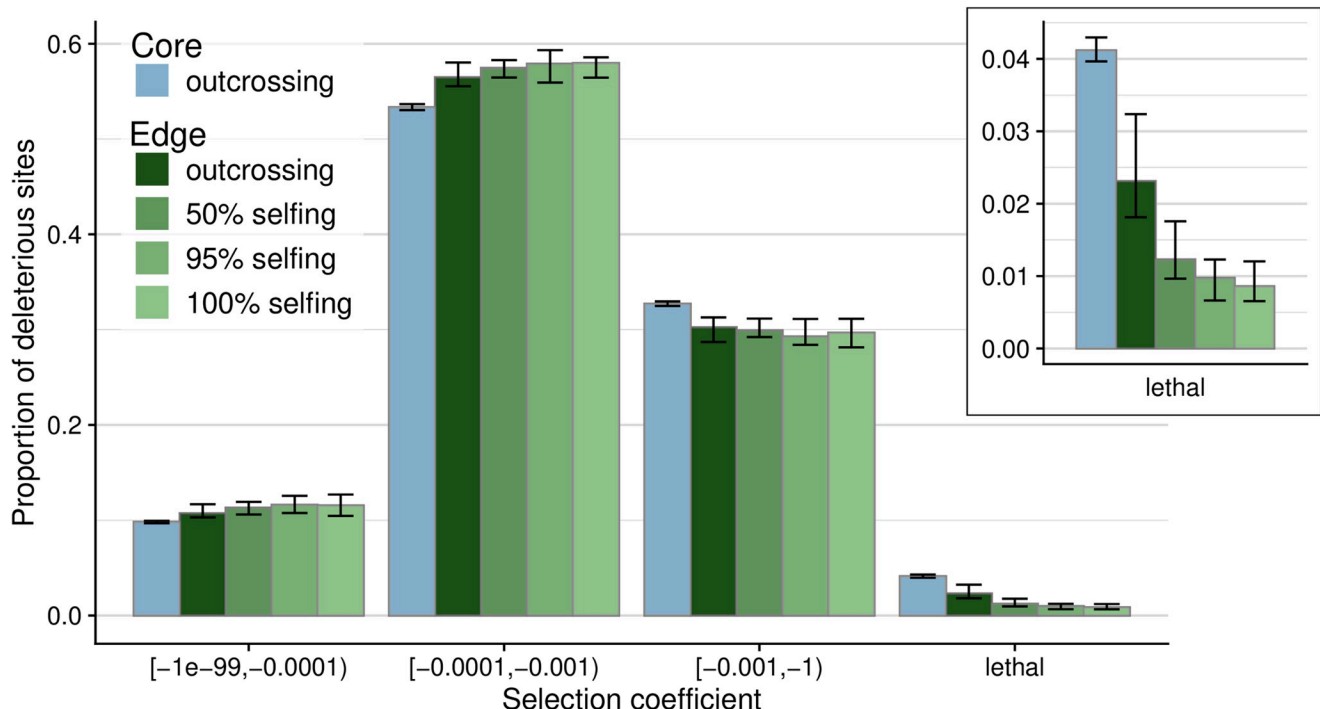

**Fig 3. The observed distribution of selection coefficients from simulations at the end of expansion.** The core deme (blue) is compared to edge demes (green) for obligate outcrossing (darkest color) versus higher selfing rates (lighter colors). Error bars indicate 0.05 and 0.95-quantiles across the 20 simulation replicates. The inset panel emphasizes the degree to which the proportion of sites in the lethal category changes over mating system scenario from core to edge.

in 527 individuals at high resolution, with a particular focus on our densest sampling across the expansion axis from central Italy into the western Alps.

Population structure results showed expected clustering by regions (S6 Fig), matching the geography of sampled populations from Abruzzo in southern Italy, the Apuan Alps in northern Italy, the French Alps, and the Swiss Alps (Fig 4A). Previously sampled individuals from Italy, France, and Switzerland that we combined with our newly sampled individuals also consistently clustered within the same geographic regions. Samples of fewer individuals from more widely across Europe showed reasonable structuring among Greek, Spanish, and Scandinavian populations. Our inferred rooted maximum likelihood phylogenetic tree provided evidence supporting the colonization of *A. alpina* into the western Swiss Alps from the Italian peninsula via the French Alps (S7 Fig). Using the Greek population ('VI') as outgroup, we found that the Abruzzo individuals form a single monophyletic clade, separated from a clade containing Apuan, French, and Swiss Alp populations. French and Swiss populations then share a more recent common ancestor since the split from the Apuan Alps, and furthermore, individuals from Switzerland form a monophyletic clade derived from the French populations. To further corroborate this expansion history, we also estimated split times between pairwise population combinations using two-dimensional site frequency spectra in `dadi`. Split times largely agreed with findings from our phylogeny (S8 Fig, S1 Table). The most recent split was estimated to be between populations of French and Swiss origin, and more ancient split times were estimated between populations separated by larger geographic distances, e.g., Apuan-Swiss, Abruzzo-Swiss and Abruzzo-France. Interestingly, some pairwise split times estimated between Abruzzo-French populations were estimated to be more recent than inferred split

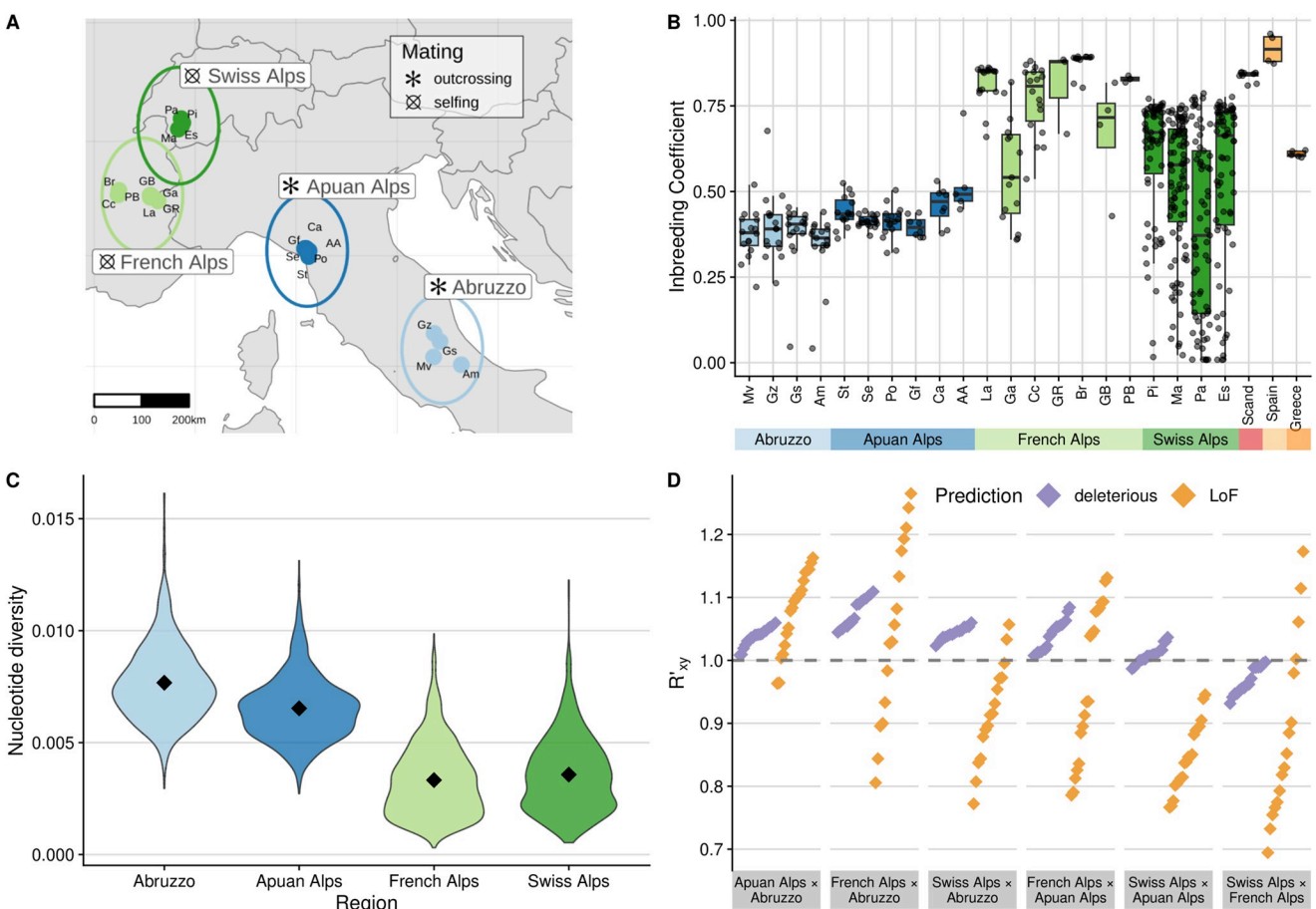

**Fig 4.** Sampling sites of *A. alpina* in the Italian-Alpine expansion zone (with mating types as published in [57, 59] and map drawn using the world dataset in the R package maps [65]) (A). Inbreeding coefficients for individuals across sampled populations, including Spain (selfing), Scandinavia (selfing), and Greece (outcrossing, [58]) (B). The distribution of nucleotide diversity estimated for Italian and alpine populations (C), with diamonds indicating group means. $R'_{xy}$ values for predicted deleterious (nonsense and missense variants; purple) and LoF (orange) loci are shown for each pairwise population comparison and sorted by increasing order within each region (D). Confidence intervals are smaller than point sizes and thus not shown. $R'_{xy} > 1$ indicates an accumulation of derived alleles at deleterious or LoF sites relative to neutral sites, while $R'_{xy} < 1$ indicates a deficit.

times between Apuan-French populations, indicating greater divergence in the Apuan populations. Speculatively, this could be indicative of unstudied complexities in the demographic history within the Apuan Alps that may have limited gene flow to a greater degree since the expansion from Italy into the western Alps. Inference of the one-dimensional demographic histories of the Italian and alpine populations showed a history of population bottlenecks and recovery consistent with the northward expansion following the last glacial maximum and the shift to selfing in France and Switzerland (see S9 Fig).

To reveal how expansion and self-fertilization impact key diversity parameters, we calculated individual inbreeding coefficients ($F$) and nucleotide diversity ($\pi$). We found the highest inbreeding coefficients in mixed mating and highly selfing populations outside of Italy, and reduced inbreeding in the Apuan Alps and Abruzzo (means for Swiss, French, Apuan Alps and Abruzzo, respectively: $\overline{F} = 0.51, 0.75, 0.43, 0.37$). The highest overall inbreeding coefficients were estimated in populations from Spain ($\overline{F} = 0.96$) and France (Br, $\overline{F} = 0.89$, Fig 4B). Swiss populations had the greatest standard deviation (Pa, $SD(F) = 0.26$), and Italian

populations had the lowest mean value (Am, $\overline{F} = 0.34$). For nucleotide diversity, we found high values in the Abruzzo region of Italy ($\overline{\pi}_s = 0.00767$). Genetic diversity reduced when moving north to the French Alps ($\overline{\pi}_s = 0.00333$) and Switzerland ($\overline{\pi}_s = 0.00357$, Fig 4C).

We calculated $R'_{xy}$ to assess the accumulation of derived deleterious alleles, using 270,889 SNPs annotated as deleterious (nonsense and missense variants) and 2380 as more severe loss of function (LoF) variants, classified by SNPeff [62]. $R'_{xy}$ is a pairwise statistic that compares the count of derived alleles found in one population relative to another, and avoids reference bias introduced by branch shortening [63]. $R'_{xy} > 1$ indicates that population X has more derived alleles of a given class than population Y relative to the neutral expectation, while $R'_{xy} < 1$ would indicate fewer derived alleles in population X. For deleterious sites we found that alpine populations had more derived alleles compared to Italian populations (Fig 4D), indicating an increase in genetic load from south to north. Within the Alps, all Swiss populations had reduced derived allele frequencies compared to France, while relative to the Apuan Alps in northern Italy, few Swiss populations exhibited reduced derived allele counts, suggesting that Swiss populations have purged expansion load to a greater degree than French populations. For LoF loci, signals of both purging and accumulation were detectable. Some pairwise population comparisons showed an increase in number of LoF alleles from south to north (e.g., nearly all Apuan × Abruzzo comparisons), while others showed mixed results, depending on the focal populations (e.g., French × Apuan, Swiss × French). All population comparisons of Swiss Alps × Apuan Alps showed reduced LoF allele counts, again suggesting particularly strong purging in the Swiss Alps.

We next used the SNPs with variants annotated as putatively deleterious to examine the accumulation of genetic load in our expanded populations. We assess both additive genetic load and recessive genetic load in our populations, as previous theoretical and empirical results show that range expansions are expected to lead to an increase in recessive load and a constant level of additive load [18, 26]. The additive load model counts deleterious alleles per individual (since alleles act additively), while the recessive model instead counts loci that are homozygous (since only homozygous recessive alleles impact fitness). Using our simulations, we evaluated how each of these models performed at predicting fitness with a simple linear model. The correlation between per-population mean fitness and load prediction was stronger for the recessive model ($R^2 = 0.82$, $P < 0.001$) than the additive model ($R^2 = 0.10$, $P < 0.001$, S10 Fig). The additive model also predicted fitness more poorly in a supplemental set of simulations using only fully additive mutations (see S11 Fig). In our empirical dataset we found a markedly increased recessive load in expanded, selfing populations from France and Switzerland, as compared to core Italian populations, indicative of expansion load (Fig 5A). Furthermore, the additive load results showed a notable decrease for expanded selfing alpine populations (Fig 5B), a departure from the expectation of a constant level of additive load during expansion [18], therefore indicative of purging.

To further understand the mutational burden within our *A. alpina* populations, we estimated the distribution of fitness effects of new mutations (DFE) using fitdadi [64], which corrects for demographic history by first fitting a best demographic model to the data. Our inferred demographic model (S9 Fig) matched well to the expansion history of these populations, showing recent bottlenecks in expanded Alpine populations, roughly corresponding to estimates of split times and the phylogeny (S7 and S8 Figs). fitdadi surprisingly estimated a similar DFE across all of our sampled populations (S12 Fig), with a large proportion of strongly deleterious sites at or above 60%, around 20% of sites in the weakest selection class, and approximately 5% in each of the two intermediate selection classes. The proportions varied only marginally across core Italian populations as well as across expanded French and Swiss

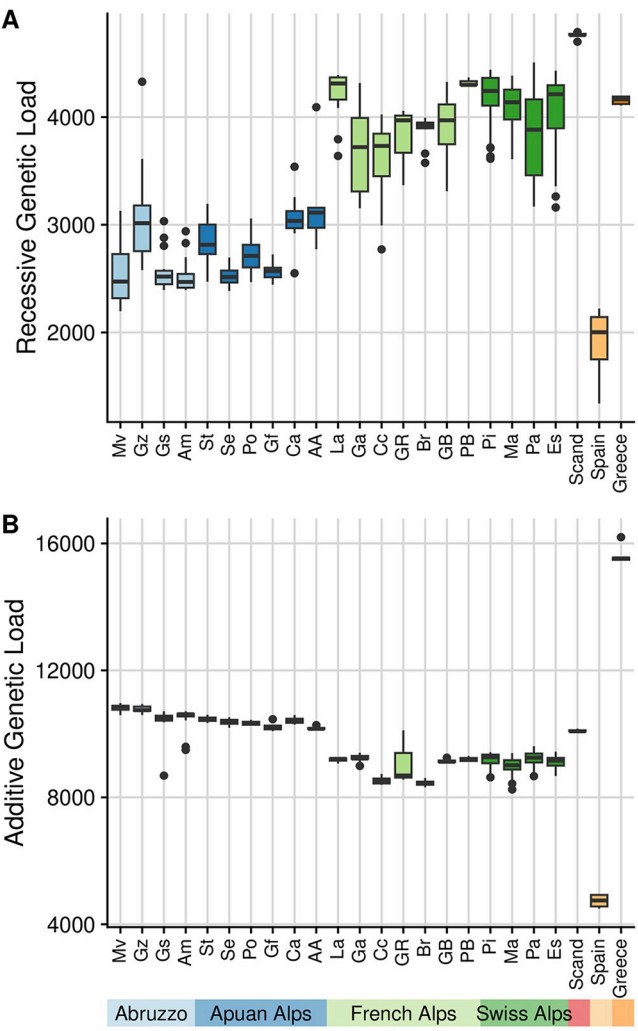

**Fig 5.** Genetic load in *A. alpina* populations as inferred from counts of homozygous deleterious loci, assuming all deleterious mutations act recessively ('recessive model') (A) versus counts of deleterious alleles, assuming all deleterious mutations act additively ('additive model') (B). Loci are classified as putatively deleterious by SNPeff (see Methods).

populations. We additionally examined the fixation of deleterious alleles across our populations, within classes of neutral, deleterious, or LoF sites (S13 Fig). Fixation of all sites increased from Italy in the south to France in the north, but then decreased from France to Switzerland, reminiscent of our $R'_{xy}$ results suggesting more purging in Swiss populations.

## Discussion

In this study, we investigated the impact of selfing on the dynamics of genetic load accumulation during a species range expansion and the genomic signatures resulting from this. Studying range expansions in plant species offers unique insights into the combination of mating system evolution combined with the evolutionary processes occurring during species range expansions. By comparing simulations with and without selfing, we disentangle the reduction in $N_e$ at expanding fronts due to serial founder events from reductions in $N_e$ due to self-fertilization.

We then compared our expectations for the impact of selfing to empirical data of natural populations which underwent both a range expansion and a mating system shift. Because selfing also reduces the effective recombination rate within populations, in addition to reducing genetic diversity, it is expected to be generally maladaptive for evolution and adaptation. However, conditions at the expanding edge of a species range may particularly favor the evolution of selfing mating systems. The compounded effects of reduced diversity due to selfing at range edges may even provide an additional benefit of purging homozygous recessive deleterious mutations. Overall, we find that expansion load accrued from a demographic history of range expansion dominates over the potential effect of purging from selfing. Yet, both in simulations and empirically we find evidence for purging deleterious load, substantiating the hypothesis of selfing providing an evolutionary advantage during range expansion.

## Mating assurance as an advantage of selfing during range expansion

Though not our primary investigation, our simulation results confirm the hypothesis that selfing provides reproductive assurance [7] and leads to faster spread over geographic space. Despite similar losses in fitness from core to range edge for both outcrossers and all selfing rate scenarios, selfers still colonized the landscape faster than outcrossers. This result adds to the general prediction of Baker's Law, that selfing may be advantageous in mate-limited environments [49]. Though, interestingly, previous work in *Campanula americana* found no change in pollinator availability across a cline in mating system, suggesting that evolutionary and genetic factors rather than ecological factors such as mate availability drive the evolution of selfing across species ranges [34, 66].

Previous theoretical results predict that speed of expansion can also play a role in the severity of expansion load accumulated, since the effective bottlenecks imposed during founding can occur over more or fewer generations [20]. The speeds observed from our simulated expansions, however, all fall within the realm of predicted severe load accumulation, and our fitness results support this, given that outcrossers and selfers accumulate similar magnitudes of expansion load despite the additional number of generations required by the outcrossers to cross the simulated landscape. Since our *A. alpina* dataset does not have equivalently expanded outcrossing populations to compare to expanded selfers, nor a generation time suitable for experimental evolution studies, we cannot empirically investigate how selfing may have allowed for faster colonization. To fully understand the benefits of reproductive assurance from selfing, it may be fruitful for future empirical studies to focus on organisms with well-documented expansion times and variable mating system shifts over different expansion axes. It could also be insightful to take advantage of laboratory experiments with expansions of outcrossers versus selfers under controlled conditions, for example using mixed mating species of *Caenorhabditis*.

## Purging as an evolutionary advantage of selfing during range expansion

A potential major benefit of selfing is the opportunity for purging due to increased homozygosity. Theory predicts that increased homozygosity should lead to efficient removal or reduction of lethal mutations [67, 68], but our simulation results show that expansion load always accumulates at similar levels at range edges, regardless of selfing rate and with equivalent severity to outcrossers. Only in an additional set of simulations of fully additive mutations increased selfing showed a trend of further improving fitness after an expansion (S1(C) Fig), however, we believe that such a mutational model of fully additive mutations is unlikely to accurately represent nature [69, 70].

Our results suggest that purging due to selfing offers no additional benefit in terms of overall population fitness during species range expansion. However, when looking at the distribution of effect sizes for variants segregating within populations, we detect significant effects of purging unique to selfers whereby lethal-effect alleles are successfully and rapidly removed from the population. Purging was most pronounced in obligate selfers, where within only 30–150 generations lethal alleles are removed from the population and remain at low levels for the remainder of the simulations (Fig 2B). Examining the distribution of mutational effect sizes at the end of the simulations also shows that selfers exhibited major reductions in lethal alleles (Fig 3), to a much greater degree than the reduction of lethals obtained by outcrossers. Purging does not, however, allow expanded populations to escape the burden inflicted by expansion load and their demographic past. Load still accumulates in all population expansions regardless of mating system, but how this load is expressed in terms of number and effect size of variants differs among mating systems.

### Genomic signatures of expansion load with purging

In our empirical *A. alpina* results we found similar signatures of both load accumulation and genetic purging in expanded populations. The recessive load model indicates that French and Swiss expanded populations have accumulated genetic load, through higher counts of putatively deleterious sites. The additive model shows a decrease of equivalent magnitude in deleterious allele counts for our expanded populations. This suggests that negative selection has purged some diversity from these populations, since otherwise allele counts are predicted to remain at constant levels during range expansions if only genetic drift is acting and not selection [18]. Our simulation tests of the additive and recessive load models show the recessive model to better predict fitness. We only use our estimations of load from the recessive and additive models to show relative changes from refugia to expanded populations, but this model fit result is reassuring given the difficulties of inferring load under complex evolutionary scenarios [71, 72].

Our $R'_{xy}$ results provide additional support for both the accumulation of expansion load and its subsequent purging. The indication of greater purging in more severe LoF variants relative to deleterious variants, i.e., missense and nonsense, matches our expectations from simulated DFE results where lethal mutations are the class of variants purged most efficiently by selfing. The additional purging even in the less severe deleterious class of variants evidenced by Swiss versus French Alp and Swiss versus some Apuan Alp populations potentially suggests that Swiss populations may self-fertilize to an even greater extent than French populations and thus also purge to a greater extent and across variants spanning a wider range of selection coefficients. The mixed evidence of purging versus accumulation for some LoF variants could reflect several potential processes. Variants classified as LoF are very few and because of their potentially high selection coefficients may respond much more quickly to selection, likely being already removed by selection prior to expansion and thereby looking like an accumulation. Or alternatively these may contribute to expansion load because purging has not been sufficiently successful to yet remove them. Across all population comparisons we still overall see a stronger effect of purging than of accumulation for these LoF variants.

Our observation that purging due to selfing during range expansions leads to different distributions of allelic effects contributing to expansion load has potentially interesting implications. Expansion load in more highly selfing populations consists of a greater proportions of small- and intermediate-effect deleterious variants, as shown from our simulation DFE analyses and also as described above in our $R'_{xy}$ results. However, our empirical DFE inferences only detected minor trends of reduced proportions of sites in the most deleterious class for

expanded Swiss and French populations relative to outcrossing core Italian populations (S12 Fig), but did corroborate the bimodal-shaped DFE reported in [58]. Whether this reflects true minor differences in the DFEs among these populations, or a lack of proper inferential ability is difficult to know. DFE inferences are known to have variable estimation accuracy depending on selfing rate and the degree of linked selection [73] or simply due to different histories of demographic change [74, 75]. Previous simulation studies have highlighted how small effect variants are much more difficult to purge [45, 76], which supports our results that expansion load clearly still accumulates despite the presence of selfing and shifts to consist mainly of these smaller-effect variants. Previous empirical work has additionally identified patterns reflecting less efficient selection against weakly deleterious additive variants and purging of strongly deleterious recessive variants in the context of mating system evolution [77] or demographic change combined with mating system evolution [58]. Mutational effect sizes and distributions are thus likely a major factor for the evolution of mating systems [74, 78]. The manner in which this genetic architecture underlying expressed load differs across mating systems may thus have important impacts on how selection and recombination interact as populations adapt in the future [79].

Previous work in *A. alpina* has also evidenced increased load with high selfing and bottlenecks in Scandinavian populations [58], however we highlight previously unidentified evidence for purging of strongly deleterious alleles in intermediate to highly selfing continental populations within the French and Swiss Alps, in addition to expansion load still incurred. Our results are also similar to those found in other plant range expansions where selfing is observed at the range edge. Notably, this is the case in *A. alpina*'s close relative *Arabidopsis lyrata* [30], as well as in *Mercurialis annua* and *Campanula americana* where expansion load has been indicated [29, 34]. Purging is a well-known process from a theoretical point of view [43, 74], but evidence of purging in natural populations is mixed [80]. Purging has been evidenced in natural selfing populations [41, 44, 77] but never directly studied during range expansions. Our study has uniquely identified signatures of purging due to selfing during a species range expansion. Future studies will still greatly benefit from direct estimates of fitness through crosses and common garden studies to better understand the true impacts of load accumulation on wild phenotypes.

Our simulations indicate that the loss of diversity at range fronts can be recovered after the expansion front has passed, once migration and population growth allow for increased efficiency of selection in larger and more diverse populations, as previously described in [20]. A novel insight from our results is that this recovery is much slower for selfing populations, supporting the widely-held idea that selfing should only be favored at range edges and that outcrossing may replace selfing after a range expansion has occurred. Encinas-Viso *et al.* [32], a simulation study investigating when selfing is favored to evolve, showed that outcrossing individuals will outcompete selfers once the expansion edge has passed, unless recombination rates are sufficiently high. Whether any of our sampled alpine populations have also began shifting back towards increased outcrossing is currently unknown and an avenue of investigation which will be interesting to pursue in the future. Given that our empirical populations still exhibit signatures of genetic load is then also interesting in light of these expectations, since our populations are no longer actively expanding edge populations and thus have had several generations over which recovery from expansion load should have begun.

## Caveats and future directions

Population genetic simulations help us to better understand interactions of effects that are difficult to assess or disentangle in empirical populations. Here, we have only explored a finite

parameter space and constrained our simulations to simplified demographic models. Our simulated range expansions occur across 1-dimensional transects, and while this should be a good approximation for range expansion along a narrow two-dimensional corridor [16], as is relevant for *A. alpina* as well as other plants constricted to mountain ridges or along valleys, incorporating 2-dimensional landscapes could provide interesting results where we might expect lateral gene flow to increase genetic diversity and potentially reduce the benefit of purging that is otherwise enhanced when homozygosity reaches extreme values. Since we were only interested in the eventual signatures resulting from selfing evolution during a range expansion, we modeled the loss of self-incompatibility as a sudden shift in the probability of selfing at one location on the landscape. However, in nature, the shift to self-fertilization is expected to occur gradually over time, e.g, due to a reduction in S-allele diversity [32, 81–83]. Even with our sudden evolution of selfing imposed in the middle of the landscape, we expect that the same observed qualitative results of purging strong-effect recessive deleterious alleles and loss of heterozygosity would still occur, just more gradually through time. The intermediate rate of selfing we tested could also be considered an earlier transitional state of a species range expansion on its way to evolving higher selfing rates. In a gradual shift to selfing, initial S-allele diversity would be reduced but outcrossing still frequent, and intermediate selfing rates would be a transient state as populations shift to higher selfing and faster expansion.

While we focused on the speed and purging benefits of selfing during a range expansion, we did not address a potential third factor impacting expansions: the necessity to locally adapt to unfamiliar environments. Populations must often adapt to novel or fluctuating environments during expansion, e.g., during glacial cycles [84] or as soil conditions change over altitude or photoperiod conditions change over latitude. Adaptation requires sufficient genetic variation to match the local environment sufficiently for population growth to be sustainable [85, 86]. For populations that expand to follow an environment they are already adapted to, this difficulty is less relevant. For example, species expanding post-glaciation are believed to have followed the receding ice sheets as suitable habitat that they were pre-adapted to was slowly revealed. However, it is still likely that some aspect of environmental conditions are always novel as organisms move over space, necessitating some level of adaptation. Our results importantly highlight how the DFE is expected to differ among outcrossed versus selfed expanding populations, creating contrasting genetic architectures within the genome. Differences in genetic architectures, i.e., few large-effect or many small-effect loci, for both adaptive and maladaptive sites have the potential to behave differently under different linkage and recombination scenarios, and are thus likely to interact with adaptation over changing landscapes, resulting in different adaptive potentials among populations. In the future, as anthropogenically-induced climate change causes more rapid changes across the landscape, the likelihood of being able to track moving environmental optima is expected to become more difficult, necessitating more rapid adaptation and emphasizing the importance of studying range expansions and shifts and the evolutionary processes involved.

## Conclusions

The concurrence between our simulated and empirical results gives striking insights into the interactions of demographic change due to range expansion with evolution of the mating system to self-fertilization. Range expansions are known to increase genetic drift and fixation of deleterious alleles, reducing fitness as a consequence. Self-fertilization further reduces $N_e$, which allows for a higher rate of fixation of weaker deleterious mutations compared to outcrossers. However, as predicted by [43] this process can also allow for short-term purging. We investigated whether this purging is realized during species range expansions and if selfing can

thus be beneficial in this evolutionary context. We described two significant factors in our simulations: first, the purging of lethal alleles is indeed observed in selfing populations, and second, this purging is not sufficient to prevent the fitness loss incurred by expansion load. Weak effect mutations accumulate to a larger extent in selfers due to the range expansion, leaving a visible signature in the DFE. Furthermore, in natural populations of *A. alpina*, we see consistent effects of purging as well as load accumulation despite the evolution of selfing. Together, this demonstrates that self-fertilization can alter the signature of genetic load in expanded populations, and identifies purging as an additional benefit of selfing along with reproductive assurance. Future studies in empirical systems will hopefully be able to distinguish expanded outcrossing versus expanded selfing populations to further validate our results, as much remains to be learned of the interaction between mating system evolution and demographic history of populations. Improved understanding of these important processes will be vital for further insight into how natural populations will (or will not) be able to disperse and adapt in the face of global climate change and anthropogenic forces experienced in natural habitats.

## Material and methods

We conduct simulations of a species range expansion and compare to an empirical dataset from the plant *A. alpina* to understand the dynamics of purging and mutation load accumulation in a system where self-fertilization has evolved. To understand whether selfing acts as an evolutionary advantage during expansion by purging deleterious alleles that otherwise accumulate, we focus on tracking genetic load in both simulated and empirical data. Though simplified from reality, our simulations have the important advantage of knowing true fitness and mutational effects within every individual to best understand the dynamics of load accumulation and purging during range expansion.

### Simulations

To simulate a range expansion with a shift in mating system we conducted individual-based, forward time simulations, using a non-Wright-Fisher model in SLiM v3.7.1 [60]. We modeled the range expansion across a one-dimensional, linear landscape of 50 demes with a stepping-stone migration model (Fig 1A). Each simulation started with a single initial core deme populated with individuals that then underwent repeated bottlenecks and founder events as they colonized the remaining empty 49 demes. The core population was initiated at carrying capacity $K = 5000$, and prior to expanding we ran a burn-in for $4N$ generations. Generations were discrete and non-overlapping, and after the burn-in was complete we opened the landscape for expansion, introducing migration that allowed individuals to move into either adjacent deme. We defined a forward migration rate of $m = 0.05$ per generation and reflecting boundaries at the ends of the landscape in the core and deme 50. All subsequent demes outside the core had a carrying capacity of $K = 200$. Once the last deme reached carrying capacity and 100 additional generations passed, we stopped the simulation.

To test the effect of increased self-fertilization during the expansion we conducted a set of obligately outcrossing simulations to serve as a null model for range expansion without the additional impact of uni-parental inbreeding arising from selfing. We then compared to three different simulated scenarios where selfing begins halfway through the expansion, in deme 25. In demes 25–50 of these selfing simulations, we set the self-fertilization rate $\sigma$ to either 0.5, 0.95, or 1. We replicated every parameter combination 20 times for a total of 80 simulations across all three selfing rates and the obligate outcrossers. In a given deme, individuals to be selfed were chosen with probability $\sigma$ each generation. We disabled incidental selfing, meaning that outcrossing was modelled as obligate outcrossing, and outcrossing rates in facultative

selfing scenarios ($\sigma = 0.5, 0.95$) fluctuate around $1 - \sigma$, regardless of population density ("nonWF" model in SLiM). We modeled logistic population growth with a Beverton-Holt model, where the expected number of total offspring per deme for the next generation is given by $N_{t+1} = \frac{RN_t}{1+N_t/M}$, where $M = \frac{K}{R-1}$, growth rate $R = 1.2$, $N_t$ is the deme's census size in the current generation $t$, and $K$ is the carrying capacity of the focal deme. For each parent, we expected less fit individuals to produce fewer offspring and thus implemented a fecundity selection model, where the expected number of offspring for individual $i$ is approximately Poisson distributed [17].

Each individual was modeled as a diploid genome consisting of $1 \times 10^7$ base pairs (bp) with a recombination rate of $1 \times 10^{-8}$ per bp per generation. We simulated neutral, beneficial, deleterious, and lethal mutations at a per base pair mutation rate of $7 \times 10^{-8}$ per generation occurring at relative proportions of 0.25, 0.001, 0.649 and 0.1, respectively. For deleterious and beneficial mutations, selection coefficients were drawn from an exponential function with mean -0.001 or 0.01, respectively, and lethal alleles had a selection coefficient of -1. We also tested two additional DFEs shifted to either more weak effect or more strong effect deleterious variants by modifying the shape of the gamma distribution but maintaining the same mean (see S1 and S5 Figs). Dominance coefficients were set to $h = 0.3$ for beneficial and deleterious alleles, and 0.02 for lethal mutations. Individual fitness in SLiM is calculated multiplicatively across all mutations an individual possesses, as drawn from these distributions for effect size and dominance coefficient. In a supplementary set of simulations we tested for the effect of full additivity using $h = 0.5$ for non-lethal mutations. These simulated parameters for the distribution of selection and dominance coefficients reflect partial dominance of deleterious alleles and more recessive lethal alleles, as described in the literature for the current best knowledge of mutational distributions in nature [70, 87–89].

We recorded fitness and calculated summary statistics during the expansion to track the impact of demographic change in combination with selfing rates. In every deme we measured nucleotide diversity for neutral variants, $\pi$, mean observed heterozygosity along the genome, $H$, counts of lethal and deleterious alleles and recorded mean fitness, $\overline{\omega}$, every five generations. This allowed us to compare changes in fitness and allele counts over time, contrasting them with the same statistic 100 generation in the past. We also examined changes in these summary statistics in specific locations across the landscape during and after the expansion had completed: the core (deme 1), the deme prior to the mating shift (deme 24), the deme ten demes past the facultative mating shift (deme 35, to avoid effects of migration from outcrossers), and the end of the landscape (deme 50). We characterized the composition of load in core and edge populations after the expansion by examining the realized distribution of selection coefficients. To do this, we categorized selection coefficients in four discrete bins ($s \in \{(-0.0001, 0), (-0.001, -0.0001], (-1, -0.001], -1\}$). To further characterize load, we calculated the proportion of fixed deleterious alleles, and applied models often used to compare approximated genetic load in empirical populations [71] to our simulated data: we estimated additive load by counting the total number of deleterious alleles per individual, assuming $h = 0.5$, and recessive load by counting the total number of homozygous deleterious loci per individual, assuming $h = 0$. We then compared these values with realized fitness, all of which are known for the simulations.

### *Arabis alpina* dataset

We compared our theoretical results to an empirical dataset of *A. alpina* by combining publicly available data [58, 61] with newly sampled and sequenced genomes. Our dataset focused on sampling four regions with four populations each, consisting of 15–18 individuals. One

exception is northern Italy where two nearby populations (Ca & Gf) of 8 individuals each contributed to five total populations from the region. Sampling spans the range expansion from southern Italy north into the French and Swiss Alps and capturing the transition in mating system from outcrossing to selfing. We collected leaf tissue on silica gel from 198 wild *A. alpina* plants in the Apennine Mountains in central Italy, the Apuan Alps in northern Italy, and the western Alps in France during the summer of June 2021. We extracted DNA with the Qiagen DNeasy Plant Mini Kit (Qiagen, Inc., Valencia, CA, USA) and constructed libraries using Illumina TruSeq DNA PCR-Free (Illumina, San Diego, CA, USA) or Illumina DNA Prep, and sequenced on a Illumina NovaSeq 6000 (paired-end). All sampled individuals are described in more detail in S1 Data and are available publicly at NCBI SRA accession PRJNA773763. We combined this dataset with previously published *A. alpina* short-read genomes of 306 individuals sampled from Switzerland [61] and 36 sampled widely across Europe [58]. For quality control of the reads, we used FastQC (http://www.bioinformatics.babraham.ac.uk/projects/fastqc) and MultiQC [90]. We trimmed reads using trimmomatic 0.39 [91] and aligned them to the *A. alpina* reference genome ([92], version 5.1, http://www.arabis-alpina.org/refseq.html) using `bwa mem` 0.7.17 [93]. To remove PCR duplicates, we used Picard Tools `MarkDuplicates` Version 2.23.8 [94]. We calculated coverage for the whole dataset with mosdepth [95], averaging at 13.98 (18.61 for new samples, S1 Data). We called variant and invariant sites using `freebayes` 1.3.2 [96]. Additional filters were applied in bcftools [97], retaining only sites with a maximum missing fraction of 0.2, and removing any variant sites with estimated probability of not being polymorphic less than phred 20 (QUAL>=20). Finally, we removed 13 individuals with greater than 30% missing calls or low coverage (Br22, Br06, Cc05, St15, Am01, Br18, Br24, Pa9, Pi9, Pi95, Pi40, Ma97). The final dataset combined had 3,179,432 SNPs, with 43,268,666 invariant sites for 527 individuals from 31 populations, which includes 191 individuals of the 17 newly sampled populations in the Italy-Alps expansion zone.

## Population genetic analyses

We inferred the ancestral state of alleles using the close relative of *A. alpina*, *Arabis montbretiana*, by aligning the reference sequences of *A. alpina* with *A. montbretiana* [98] using `last` [99]. To confirm that our samples from across Europe matched the expected population structuring based on known demographic history, we ran `admixture` v1.3.0 [100] from $K = 2$ to $K = 15$ on the full sample set but with SNPs pruned for LD using `bcftools +prune` ([97], $R^2$ cutoff 0.3 in a window of 1000 sites). We used the same dataset but additionally subsampled to a maximum 10 individuals to further analyse population relatedness and history of the Italian-Alpine expansion axis using `RAxML` 8.2.12 [101] to construct a maximum likelihood phylogenetic tree (see S7 Fig for more details). We calculated nucleotide diversity per population in 1Mbp windows using `pixy` ([102], version 1.2.6.beta1, 10.5281/zenodo.6032358) and inbreeding coefficients for each individual with `ngsF` (6 iterations, [103]). To format the input file for `ngsF`, we randomly sampled 100,000 biallelic SNPs and extracted genotype likelihoods using bcftools [97].

To predict deleterious alleles we annotated the variant calls with SNPeff [62]. SNPeff estimates how deleterious a variant may be based on whether its mutation causes an amino acid change, at varying levels of importance [62]. We used the SNPeff categories "nonsense" and "missense" as the definition for derived deleterious mutations, "none" and "silent" annotations were used as neutral predictions and "LoF" annotations as loss-of-function mutations ("LoF") after running the program with the `−formatEff` option.

With these SNPeff annotations, we estimated four different statistics to infer genetic load: $R'_{xy}$ [63], recessive and additive genomic load (by counting homozygous deleterious loci or

deleterious alleles, see simulations), and inferred the DFE [64]. We calculated $R'_{xy}$ as described in [63] with $R_{xy}$ for derived allele counts of LoF or deleterious sites over $R_{xy}^{normalization}$ for synonymous sites to avoid reference bias. We estimated jackknife confidence intervals using pseudo values from 100 contiguous blocks and assuming normal distributed values. For recessive and additive genomic load, we used the SNPeff predictions and the same assumptions as the load approximations described in the simulation section, by counting either the total number of homozygous deleterious loci or derived alleles at deleterious sites. Finally, we estimated the empirical DFE for every population using `fitdadi` [64] and `dadi` [104] (python 3.8.12) in 100 replicated runs (see S9 and S12 Figs for supplementary methods and results).

Statistical analyses were conducted in R v.4.1.3 [105], unless otherwise specified.

## Supporting information

**S1 Data.** Description of *A. alpina* samples.
(CSV)

**S2 Data.** Demographic inference output from dadi.
(CSV)

**S1 Table.** Split times, in generations, for pairwise comparisons between populations from different regions. We calculated split times using dadi by fitting a bottlegrowth model with a population split. Comparisons between populations within the same region are omitted.
(PDF)

**S1 Fig.** Similar to Fig 1D, we assessed relative fitness for additional simulations where non-lethal mutations where drawn from gamma distributions with with the same mean as our original parameter set ($\bar{s} = -0.001$) but using shape parameters which shift the distribution to contain a higher proportion of weak-effect variants ($\alpha = 0.5$) (A), or to contain a high proportion of large-effect variants ($\alpha = 2$) (B). Lastly we also compared to a case using our original DFE shape (exponential distribution with $\bar{s} = -0.001$) but instead with additive mutations ($h = 0.5$) for all non-lethal variants (C). Other parameters remained as described in the main text.
(PDF)

**S2 Fig.** The proportion of sites fixed for deleterious alleles (A), the mean counts of deleterious loci (B), and the mean counts of deleterious alleles (C), all assessed at the end of the simulations for core and edge populations across selfing rates. Whiskers indicate the 1.5 interquartile range.
(PDF)

**S3 Fig.** Trajectories for the mean observed heterozygosity over relative time, as described in Fig 2A, but now including all simulated selfing rates.
(PNG)

**S4 Fig.** Trajectories for the mean count of lethal alleles over relative time, as described in Fig 2B, but now including all simulated selfing rates.
(PNG)

**S5 Fig.** Using the same parameter sets described in S1 Fig and similar to Fig 3, we compared the distribution of selection coefficients after the expansion. Insets emphasize strong reductions of proportions of lethal alleles with increased selfing, regardless of DFE or dominance coefficient parameterization. Error bars indicate 0.05 and 0.95-quantiles across the 20

simulation replicates within parameter combination.
(PDF)

**S6 Fig.** Results from $K = 2$ to $K = 15$ from admixture analyses run on the combined empirical dataset across Europe. The lowest CV error is for $K = 14$, however it is most useful to compare the populations structure across values of $K$ to see how well this matches known geography and demographic history of the populations. We observe clean distinctions among our geographic regions sampled (indicated above the bar plots), with evidence for some gene flow across geographic space as one observes higher $K$ values.
(PDF)

**S7 Fig.** RAxML phylogenetic tree calculated using the rapid bootstrap analysis with 1000 replicates and search for bestscoring ML tree option in RAxML (option `-f a`). We used 'GTRGAMMA' as the substitution model, and *A. alpina* individuals from Greece (population 'VI') as outgroup. For computational reasons we randomly subsampled to a maximum of 10 individuals per population and used the same pruned SNP dataset as in the admixture analysis.
(PDF)

**S8 Fig.** The range from minimum to maximum for split times between pairwise populations across regions is shown. Split times were estimated using dadi, and similar to the one-population demographic estimates (S9 Fig), we used 100 replicates and retained the best fitting replicate run based on log-likelihood. Estimates for every individual pairwise population across regions are listed in S1 Table.
(PDF)

**S9 Fig.** We inferred the demographic history of each of our newly sampled populations of *A. alpina* along with the densely sampled Swiss populations using dadi. This is a necessary step to account for the demography when inferring the DFE with fitdadi. This also allowed us to confirm if this newly inferred demographic history is consistent with past studies in *A. alpina*. The best-fitting models for our populations, based on AIC, were "bottlegrowth" models, indicating a past bottleneck followed by exponential growth (Es, Ca, Gf, Gz, Po), three epoch models, indicating a bottleneck followed by a sudden size change (Pa, Pi, Ma, Am, Br, Cc, Ga, Gs, La, Mv, Se), and the standard neutral model (St). Populations St and Gz were the only instances where competing models fitted approximately equally well (see S2 Data), therefore results for these population should be interpreted with caution. With the exception of Es, all Alpine populations best fit to three epoch models. Central Italian populations (light blue) show the most historic bottlenecks and the largest ancestral populations sizes. This is consistent with this region of highly outcrossing plants being subject to the last glacial maximum. Northern Italian populations (dark blue) show more recent bottlenecks and reduced ancestral sizes relative to central Italy, potentially reflecting their expansion northward. French and Swiss Alpine populations both showed the most recent bottlenecks and the smallest historic population sizes, consistent with both their shift to selfing and their more recent range expansion. Depleted genetic diversity along the axis of an expanding species range is expected, as is decreased $N_e$ due to inbreeding and thus loss of diversity. These demographic inferences thus match our understanding of both the mating system shift and the range expansion that these populations experienced.
(PDF)

**S10 Fig.** Observed (known) mean fitness from simulations for core (green), interior (orange, purple) and edge (pink) demes compared to the inverse of the count of deleterious loci (A), both after the range expansion is complete. The count of deleterious loci serves as a model for

recessive load, which we find best correlates to fitness, compared to the additive model (B), where load is predicted by counting alleles. Results are for simulations with $h = 0.3$ for non-lethal deleterious mutations.
(PDF)

**S11 Fig.** Recessive (A) and additive (B) genetic load compared with known simulated fitness to infer load when all non-lethal deleterious mutations are perfectly additive ($h = 0.5$). Data is from a supplementary set of simulations with these dominance parameters. This repeats the same analyses as S10 Fig, except now for simulations with additive mutations. This result again finds that the recessive model predicts load better ($R^2 = 0.70$, $P < 0.001$) than the additive model ($R^2 = 0.20$, $P < 0.001$).
(PDF)

**S12 Fig.** We inferred the DFE of each *A. alpina* population in the Italy-Alps expansion zone using `fitdadi` from `dadi` in python 3.8.12. We used the SNPeff annotation to construct polarized site-frequency spectra for neutral and deleterious sites after subsampling to a maximum population size of 20 individuals. To estimate demographic parameters, we tested the default single population demographic models (standard neutral model, two-epoch, growth, bottlegrowth, three-epoch) and two models accounting for inbreeding (standard neutral with inbreeding, two-epoch with inbreeding). We assumed a per base pair mutation rate of $\mu = 7 \times 10^{-9}$ per generation, ran the default optimization for 100 replicates, and selected the best fit parameters within each demographic model based on likelihood and the best fit demographic model based on AIC. For `fitdadi`, we additionally assumed $L_{ns}/L_s = 2.85$, dominance coefficient $h = 0.3$ and estimated the DFE for each model in 100 optimizations. We then chose the best-fit DFE optimization based on likelihood for each population for the previously chosen demographic model. DFE results from *A. alpina* populations across the Italian-Alpine range expansion for outcrossing populations from Abruzzo (light blue) and the Apuan Alps (dark blue) are compared to the selfing populations that have undergone range expansions into the French Alps (light green) and the Swiss Alps (dark green). We found mean proportions across all populations of 65.4% and 24.8% in the weakest and strongest selection classes, respectively. Less than 5% of sites segregated in the two intermediate selection classes. These proportions varied only marginally between core Italian populations (mean proportions 22.6% and 67.9% for weakest and strongest classes, respectively) and between expanded French and Swiss populations (means proportions 27.4% and 62.6% for weakest and strongest class).
(PNG)

**S13 Fig.** Fixation of predicted neutral (dark purple), deleterious (light purple) and loss of function (LoF, orange) sites per population. Y-axis shows the proportion of fixed sites in each focal population by allele category. We found that neutral sites fixed at the highest proportions (mean 0.505%), while LoF sites were at the smallest proportions fixed (mean 0.314%), indicative of their highly deleterious effect. French populations Br and La had the highest overall fixation proportions of any class (0.948%), while samples from the Abruzzo region had the lowest (0.228%). Swiss populations showed intermediate neutral fixation but LoF proportions similar to Italian populations.
(PDF)

## Acknowledgments

We thank Drs. Marco Andrello, Michele Di Musciano, Marta Binaghi, Paola Morini, Marco Caccianiga, Alessandro Alessandrini, Rodolfo Gentili, and Enzo Bona for essential help in

finding wild populations of *A. alpina* to sample in Italy and France. We thank Dr. Pamela Nicholson and the team at the Bern NGS Platform for sequencing and troubleshooting help as well as Ryan Gutenkunst for help troubleshooting the dadi analyses. Computation was performed in part on UBELIX (http://www.id.unibe.ch/hpc), the HPC cluster at the University of Bern. We thank Stephan Peischl for useful feedback on the manuscript and Xuejing Wang for statistical advice.

## Author Contributions

**Conceptualization:** Leo Zeitler, Christian Parisod, Kimberly J. Gilbert.

**Data curation:** Leo Zeitler, Kimberly J. Gilbert.

**Formal analysis:** Leo Zeitler, Kimberly J. Gilbert.

**Funding acquisition:** Kimberly J. Gilbert.

**Investigation:** Leo Zeitler, Kimberly J. Gilbert.

**Methodology:** Leo Zeitler, Kimberly J. Gilbert.

**Project administration:** Kimberly J. Gilbert.

**Resources:** Kimberly J. Gilbert.

**Supervision:** Christian Parisod, Kimberly J. Gilbert.

**Visualization:** Leo Zeitler, Kimberly J. Gilbert.

**Writing – original draft:** Leo Zeitler, Kimberly J. Gilbert.

**Writing – review & editing:** Leo Zeitler, Christian Parisod, Kimberly J. Gilbert.

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
