## [Decision Letter · Decision Letter 0]

31 Mar 2023

Dear Dr Gilbert,

Thank you very much for submitting your Research Article entitled 'Purging due to self-fertilization does not prevent accumulation of expansion load' to PLOS Genetics.

The manuscript was fully evaluated at the editorial level and by 3 independent peer reviewers. The reviewers appreciated the attention to an important problem, but raised some substantial concerns about the current manuscript. Based on the reviews, we will not be able to accept this version of the manuscript, but we would be willing to review a much-revised version. We cannot, of course, promise publication at that time and the manuscript will be sent to the same reviewers, but possibly different reviewers.

Should you decide to revise the manuscript for further consideration here, your revisions should address the specific points made by each reviewer. The main point of concern from the editor's perspective is the point made by reviewers 1 and 3 about the novelty of the study and being very explicit in setting up the questions in the introduction. The editors feel that the combination of the empirical work and the theoretical work is an asset. We will also require a detailed list of your responses to the review comments and a description of the changes you have made in the manuscript.

If you decide to revise the manuscript for further consideration at PLOS Genetics, please aim to resubmit within the next 60 days, unless it will take extra time to address the concerns of the reviewers, in which case we would appreciate an expected resubmission date by email to plosgenetics@plos.org.

We are sorry that we cannot be more positive about your manuscript at this stage. Please do not hesitate to contact us if you have any concerns or questions.

Yours sincerely,

Rodney Mauricio, Ph.D.

Academic Editor

PLOS Genetics

Bret Payseur

Section Editor

PLOS Genetics

Reviewer's Responses to Questions

**Comments to the Authors:**

Reviewer #1: Review of "Purging due to self-fertilization does not prevent the accumulation of the expansion load"

In this paper, the authors combine SLiMulations of expanding populations with differing mating systems, and the analysis of empirical population genomic data to investigate how mating system interacts with population expansion to determine the genetic load. I find some aspects of this work interesting, and the analyses largely strong, however, I have some trouble understanding the novelty /importance and situating this work in the broader literature. Below I outline this concern, and a few other suggestions which could improve the work.

On the whole I think that both the data and theory are promising, but neither was explored or presented in much detail and it is not clear they are sufficiently complimentary as to publish them together. If I was an author on this paper I would advocate for two papers at more modest journals - rather than packaging this all into one PLoS Genetics paper - but I am well aware that I am not an author :P

** Context / Importance / Advance **

My greatest concern with the SLiMUlations revolved around the motivation for this work. We know that both selfing and range expansion reduce the efficacy of selection on the "average excess" and increase the exposure of deleterious recessive variants allowing for more efficient purging of highly deleterious receive variants. I believe the motivation of this work was to see how these forces interacted - but it was never fully clear to me what as at stake. The result that on balance on their SLiMulations these effects largely counteract one another such that there is no effect of mating system on the genetic load on balance is interesting, but is likely sensitive to the parameters chosen (e.g. a higher or lower proportion of highly recessive and highly deleterious variants could tip the scales). Stronger motivation up front about the outstanding question being addressed and why it matters more broadly (beyond "Whether this prediction holds when a species range expansion occurs concurrently with a mating system shift has, to our knowledge, not been fully explored") would strengthen this paper.

** A related concern was the connection between the theory and data **

The paper presents this empirical case as a difference in selfing vs outcrossing. However the data seem less clear cut. Assuming the "inbreeding coefficient," F , is F_IS, the extent of (?biparental?) inbreeding in the "outcrossing" populations seem quite high (nearly 50% applying the equation F = s/(2-s)), and the selfing rates in the selfer seem to be bout 90%. So, maybe our focus should be on those parameter values? Additionally, the bottleneck in simulation seems much more extreme (in terms of a reduction in sequence variation, than that studied in nature -- compare Figure 1C to 4C).

------ Technical concerns ------

In addition to these 'big picture' concerns I had some technical questions

** Use of the "Recessive load" calculation.**

The author's claim that the recessive rather tan the additive load calculation is more appropriate for nature populations because their simulations show that this is more strongly correlated with fitness (Figures S6 and S7). While there is indeed a higher R2 here, the recessive model seems to violate the assumptions of a correlation -- namely it appears that the residual value depends on its prediction and the prediction is generally quite poor for "core" sites. I'm not sure about the best way forward here, but while the R2 is clearly lower for the additive model, it seems unbiased.

** Details of SLiMulation **

I had trouble understanding how selfing and mate limitation where baked into the SLiMulation. I know you can set a selfing rate in SLiM, but I also know that this selfing rate does not include "incidental selfing" in a randomly mating population. Did the authors add any mate limitation, if so what? If not, it seems that some incidental selfing occurred.. Anyways more details of this model would be necessary to evaluate it. Additionally, it is not clear if populations have a genetic selfing rate and it seems likely that selfing rates increase under mate limitation by e.g. geitonogamy, delayed selfing, and/or less competition between self and other pollen

UPDATE: Right before submitting this I saw the github link and found that the authors typed: initializeSLiMOptions(preventIncidentalSelfing=T), so that alleviates one concern.

** Hazards of NGS approaches **

The authors combine different sort of data (i.e. depth sequencing technology etc etc differs, see Supp1.csv), all of these issues, as well as divergence from the reference genome, can impact genotype calls, and could potentially introduce subtle biases into the analyses.

Reviewer #2: "Purging due to self-fertilization does not prevent accumulation of expansion load" (PGENETICS-D-23-00075)

In this paper, the authors conducted simulations to examine the relationship between different selfing rates and genetic load during range expansion using a stepping-stone model of migration to new demes. The authors compared core outcrossing demes to interior and edge demes that were either outcrossing, 50% selfing, 95% selfing, or 100% selfing. They compared the speed of colonization of new demes, nucleotide diversity, relative fitness, observed heterozygosity, the count of lethal alleles, the rate of fitness change over time, and the proportion of deleterious sites that fell within a range of selection coefficients from lethal to weakly deleterious. They found that selfers colonized demes more quickly, that nucleotide diversity, relative fitness, and observed heterozygosity were all reduced during range expansion and generally more for selfers than outcrossers, that the number of lethal alleles was greatly reduced for selfers compared to outcrossers, that the initial reduction in fitness was more dramatic for selfers compared to outcrossers, and that there was purging of lethal and somewhat deleterious alleles but that load of more weakly deleterious alleles did accumulate in selfers compared to outcrossers.

The authors then tested the hypothesis that they would see the same or similar results when comparing outcrossing and selfing populations of Arabis alpina. The selfing populations of this species are known to be the result of recent range expansion into the French and Swiss Alps from ancestral populations in Italy. They isolated DNA from 198 A. alpina individuals collected from selfing and outcrossing populations, used short-read sequencing, then assembled these short-read genomes and combined them with 342 existing short-read genomes for this species. They identified over 3 million SNPs in the 31 sampled populations, then used this SNP data to calculate the inbreeding coefficient and nucleotide diversity in selfers vs. outcrossers, as well as use several measures to calculate if genetic load was accumulating or being purged in selfers vs. outcrossers. They found similar results in A. alpina compared to the simulations, including that alleles with large, deleterious effects (loss of function) were purged more in selfers but not weakly deleterious alleles.

Overall, I thought this was a very interesting study, and a nice pairing of theoretical and empirical results. The majority of the paper is very well-written and clear, as well as most of the figures. However, I have a few comments:

Figure 2, Figure S2, and Figure S3: I found the color contrast of these figures to be insufficient. I really had trouble distinguishing between the green and blue, especially the lighter shades. This was the most difficult on Figure S2 and S3, where there are finer gradations in shading of green and blue used for the different selfing rates. I can see the trends, but I can’t see the difference between 50% selfing and 95% selfing, for example. I think more color contrast would improve the readability of these figures.

Methods section: What were the outcrossing rates of each of the 31 populations of A. alpina sampled for this study? I couldn’t find that information anywhere and I would like to know how the outcrossing rates of these populations compare to what was used in the simulations. Are these populations closer to 50% selfing or 95% selfing?

Results/Discussion: The authors state in the Results that there is more purging observed in the populations in the Swiss Alps compared to the French Alps. These are both regions with populations categorized as selfing. You can see these purging differences in Fig. 4D and Fig. S9, especially. But I couldn’t find any mention in the Discussion of why the authors think they are seeing these purging differences. I would like to see that discussed explicitly.

Discussion, p. 11, lines 314-320: Here the authors discuss the recessive load model versus the additive load model and that there is evidence for load accumulation of one and purging of the other. The way this is discussed it sounds as if both models are true simultaneously and I am confused. The way I understood Figure 5, Figure S6, Figure S7, and what was stated in the corresponding section of the Results, was that the recessive load model was a much better fit to the simulation results, but that both models were applied to the A. alpina data and showed contrasting results. I think I am missing something here and I need clarification in the text (and perhaps in the Figure 5 legend) so that this makes sense.

Discussion, p. 12, line 352: Currently says “Whether our sampled alpine populations populations”, but should say “Whether our sampled alpine populations”

Discussion, p. 12, lines 363-365: The authors state here that they identified purging due to selfing but don’t know of other thorough investigations in empirical systems. I assume they are referring only to purging during a range expansion because they surely can’t mean purging in general. There are certainly other studies of purging due to selfing. The work of Michelle Dudash in Mimulus guttatus comes to mind. I do think that prior work on purging due to selfing should be compared here, even if it was not explicitly testing for purging during a range expansion. I think it would add to the quality of the Discussion to have a more thorough comparison to prior empirical work.

Figure S9 legend: It reads “Y-axis shows the proportion of fixes sites in local population and allele category.” This sentence should read “The Y-axis shows the proportion of fixed sites in each local population by allele category.” Also, the last sentence says “Swiss population” where it should say “Swiss populations”.

Reviewer #3: This is my review of the article “Purging due to self-fertilization does not prevent the accumulation of expansion load”. The study addresses the effect of range expansion with or without a shift to self-fertilization on the speed of colonization and the evolution of mutational load. The paper combines results of a simulation study with molecular data on a specific plant species.

I find the general topic of the manuscript novel and highly original. It adds to the relatively young field of range dynamics and mutational load with a very meaningful contribution. Of particular value are the results of the simulation study; the empirical results presented in the paper I found less convincing. The paper is generally well written, though the Introduction lacks the clarity; it could be better structured.

General comments:

1 Abstract and Introduction. Sentences that follow each other are sometimes disconnected (e.g., Abstract, sentences 1 and 2). Some terms that are used are too unspecific (e.g., “evolutionary challenges” in Abstract; L48 “difficulties at range fronts”; L62 “adaptive measures”). Both Abstract and Introduction would gain clarity if the potential effects of a mating system shift during range expansion were split into: the ecological advantages and – if any – disadvantages, and the evolutionary advantages and disadvantages. Right now, ecological and evolutionary implications are intermingled. As a result, it remains confusing what this study addresses and which outcome gives an answer to what. I would strongly emphasize this dichotomy of ecological and evolutionary implications, from Abstract to Introduction and later in presenting results and discussing them. E.g., the study of speed of colonization targets ecological aspects of selfing. Also, I strongly recommend parallel structure, always e.g., talking of ecology first and then evolution second, such that this separation becomes very clear.

2 The simulation study is definitely the strong part of the paper. However, it is not really introduced in the Introduction; there, more emphasis is given to the empirical study system. I suggest to clearly state that simulations were done and what the goals were first, and then mention the empirical system and the goals of that one second. Introduce them both by providing similar levels of detail. What were the specific hypotheses in the two parts? Similarly, the Results section should be clearly split into two parts: outcome of simulations, outcome of empirical study. It would help to see two titles that reflect this split. Another split within those two parts should separate ecological implications of a mating system shift – speed of colonization, and evolutionary implications of a mating system shift – changes in load. Also, I recommend this split for the Discussion – simulations/empirical study + ecological/evolutionary implications. The discussion could emphasize more the novelty of the simulation results, and cite more empirical papers that have addressed mating system shift in the context of range expansion and magnitude of load (novel papers of Siberian and North American Arabidopsis).

3 Simulations. Range expansion was modelled across a one-dimensional linear landscape. I could imagine that the magnitude and effect of drift may be different (reduced) if the landscape was two-dimensional. So far, most simulation work on expansion load was on 2-dimensional landscapes. I think that authors need to address potential deviations in one way or another – by verifying their results in 2-dimensional landscapes or by comparing their predictions with those e.g., produced by Peischl et al. under similar settings.

4 I have a problem with the empirical part of the study. A first problem is a lack of information on the expansion history. The aspect of expansion is key to the research presented, and therefore, the authors need to provide data on how the expansion progressed in space in the study organism. The authors cite Tedder et al. 2015 which I checked. However, that study only showed that 3 outcrossing populations of Arabis alpina from central Italy and 3 selfing populations from the Alps fell into two separate clusters of microsatellite markers. This is no evidence that the species colonized the Alps from refugia in central Italy. The authors provide more structure results with their data, but those do not provide any insights into the past expansion history either. To learn about that, authors would need to produce some rooted population relatedness tree. Alternatively, they need to present results on demographic modelling in the main paper and provide data on split times among Italian populations and populations of the Alps that match those of glacial retreat.

5 The second problem is the estimate of load used in the paper. The authors introduce it very briefly such that it remains unclear what it really is. Also, I think it is not used often, and therefore, authors need to add other estimates that have been used e.g., in human pop. genomics or other lit. From what I read in the paper, the estimate is based on differences between populations in the count of derived alleles. Such an estimate gives a lot of weight to the many rare heterozygote variants that may contribute little to load if deleterious alleles are predominantly recessive (see Discussion in Henn et al. 2016 PNAS). I also do not understand what I should see in Fig 4D. First, the authors do not mention what each symbol stands for. Then, I see that the purple dots have similar positions in the first 5 groups but are lower in the last group; among alpine populations, the difference in counts of derived, deleterious alleles was lower than the difference in counts of neutral alleles. Based on that, authors seem to argue for a history of purging. For strongly deleterious alleles, loss of function alleles, they find that differences are regularly lower in comparisons excluding Italy-Italy, but sometimes also higher. This suggests mixed evidence of purging for highly deleterious mutations. If my interpretation is correct, this would be somewhat against the predictions based on the simulations, wouldn’t it? – All in all, this measure of load may be fine, but it should be – for reasons of comparisons and in line with the discussion in Henn et al. – be accompanied by other estimates of load (e.g., that give less emphasis on rare alleles that are mainly in the heterozygous state).

6 A third problem is the fraction of missing data across sites/variants and the fraction of missing data per individual, especially if the latter is geographically biased because of differences in coverage. Cutoffs for missing data were set very liberally. --- Missing data may be biased towards regions of the genome with higher mutation rates that also do not align well. While this may mainly increase variance in results, any geographic pattern in bias would become problematic. I recommend being more stringent, which would result in still high enough SNP numbers (now >3 Mio.).

Specific comments:

L48-50. The authors seem to have plants in mind and mention pollination. But what about other Allee effects affecting other types of organisms? Else, mention that in plants, pollination is vulnerable to an Allee effect.

L42-62. The paragraph is not as clearly written as it could be. I recommend writing in a more structured, more condensed way, mentioning the evolutionary implications that range expansions have, and then raising the theme how a shift in mating system may change predictions, and that this is what was addressed in the article.

L63-71. This paragraph (and the next) would benefit from a clearer structure, introducing the potential ecological advantages of selfing, and its evolutionary advantages/disadvantages, in the context of range expansion. I would first introduce the relevant theory and then the empirical results found so far. Or, in other words, I would e.g.., devote separate paragraphs to the theme of range expansion and mating system shift to selfing driven by selection for reproductive assurance.

L72-. An Allee effect is based on ecology. Low density or small population size lowers fitness (positive density dependence).

L77-80. Reference missing.

L159-162. What is the difference between mean counts of deleterious alleles and counts of deleterious alleles? Why the difference in outcome?

Methods. Was there a difference in average coverage (after filtering) for selfing populations?

Minor comments:

L12. A bit a weird sentence. Have not all species expanded at some point?

L15. Reproductive assurance instead of reproductive reassurance.

**Have all data underlying the figures and results presented in the manuscript been provided?**

Reviewer #1: **No: **I think this is planned for later so I am not concerned.

Reviewer #2: Yes

Reviewer #3: Yes

PLOS authors have the option to publish the peer review history of their article (what does this mean?). If published, this will include your full peer review and any attached files.

Reviewer #1: No

Reviewer #2: No

Reviewer #3: No

---

## [Decision Letter · Decision Letter 1]

25 Jul 2023

Dear Dr Gilbert,

We are pleased to inform you that your manuscript entitled "Purging due to self-fertilization does not prevent accumulation of expansion load" has been editorially accepted for publication in PLOS Genetics. Congratulations!

Yours sincerely,

Rodney Mauricio, Ph.D.

Academic Editor

PLOS Genetics

Bret Payseur

Section Editor

PLOS Genetics

Comments from the reviewers (if applicable):

We attempted to send your revised manuscript to the 3 original reviewers; the most critical reviewer accepted, the most enthusiastic reviewer, has been, unfortunately, in hospital for an extended period of time and the third reviewer was unavailable for a review. The academic editor attempted securing another reviewer, but was unable to in a reasonable amount of time. The academic editor has carefully reviewed the resubmitted manuscript along with the review from the original reviewer. I am enthusiastic about the manuscript and agree that the resubmission has addressed all the serious points raised by the original 3 reviewers and feel comfortable proceeding without additional reviews. Although the manuscript is acceptable as is, I would strongly urge the authors to consider edits in line with the reviewer's latest comments.

Reviewer's Responses to Questions

**Comments to the Authors:**

Reviewer #1: Review of "Purging due to self-fertilization does not prevent accumulation of expansion load"

This is an interesting paper and the authors did a nice job of responding to previous comments. I have a few minor follow-up suggestions / concerns.

1. Is the mating system stable? Are evolutionary transitions expected evolutionarily and tolerated ecologically?

The authors model the evolution of a genetic load for a fixed selfing rate which changes once t crosses the the threshold deme. As such, mating system does not "evolve" and may or may not be evolutionarily stable across the simulation (i.e. selfing may be favored or disfavored at any point in the range, but the manuscript is unconcerned with evolutionary stability or invasibility of a change in mating system). Maybe this is ok -- dealing with these issues is a pain. But perhaps a brief discussing of this limitation ad brief check of realism (e.g. are parameters in the rough range to disfavor selfing without mate limitation but disfavor selfing with mate limitation).

Similarly, details of the nonWF simulation where murky, but from my read I could not see how individual fitness mapped onto the population growth rate (e.g. was local extinction possible if fitness was too low?). More details here would help.

2. Concerns about the fit of the additive and recessive load models.

Perhaps this is not important, but I am still unsatisfied by the poor fit of the additive model, and the good fit, but poor diagnostics of the recessive model. This is because a violation of assumptions usually means the model is wrong, even if it fits well. I wonder if the authors could fit a model in which more deleterious alleles are more recessive.

3. I think the admixture program assumes random mating.

Something like instruct would probably be more appropriate. But I'm not sure there is a modern version of instruct capable of handling this data set, and I don't see this a s a major issue. Perhaps a brief acknowledgment of this limitation would be worthwhile.

**Have all data underlying the figures and results presented in the manuscript been provided?**

Reviewer #1: Yes

PLOS authors have the option to publish the peer review history of their article (what does this mean?). If published, this will include your full peer review and any attached files.

Reviewer #1: No

**Data Deposition**

http://datadryad.org/submit?journalID=pgenetics&manu=PGENETICS-D-23-00075R1

**Press Queries**

---

## [Editor Report · Acceptance letter]

24 Aug 2023

PGENETICS-D-23-00075R1 

Purging due to self-fertilization does not prevent accumulation of expansion load 

Dear Dr Gilbert, 

We are pleased to inform you that your manuscript entitled "Purging due to self-fertilization does not prevent accumulation of expansion load" has been formally accepted for publication in PLOS Genetics! Your manuscript is now with our production department and you will be notified of the publication date in due course.

With kind regards,

Judit Kozma

PLOS Genetics

On behalf of:
